# Second breast cancer following negative breast MRI: Analysis by interval from surgery and risk factors

**Yohan Joo, Min Jung Kim, Jung Hyun Yoon, Miribi Rho, Vivian Youngjean Park** 🄳 *

Department of Radiology, Research Institute of Radiological Science, Yonsei University College of Medicine, Seoul, Republic of Korea

* vivianpark0116@yuhs.ac

## Abstract

### Objectives

This study aims to compare outcomes following a negative surveillance MRI study by surgery-MRI interval and investigate factors associated with second breast cancers in women with a personal history of breast cancer (PHBC).

### Methods

This retrospective study included 1552 consecutive women (mean age, 53 years) with a PHBC and a negative prevalence surveillance breast MRI result between August 2014 and December 2016. The incidence and characteristics of second breast cancers were reviewed and compared according to surgery-MRI interval (< 3 years vs ≥ 3 years). Logistic regression analysis was used to investigate associations with clinical-pathologic characteristics.

### Results

Twenty-five second breast cancers occurred after negative MRI. The incidence of second breast cancers or local-regional recurrence did not significantly differ by surgery-MRI interval. The median intervals between MRI to second breast cancer detection showed no significant difference between the two groups (surgery-MRI interval <3 years vs. ≥ 3 years). Two node-positive second breast cancers were detected in the group with <3 years interval. BRCA mutation status, receipt of breast-conserving surgery, and adjuvant chemotherapy (all $p$ < .05) were significant factors associated with the development of second breast cancers.

### Conclusion

Outcomes following a negative surveillance MRI did not differ by surgery-MRI interval. BRCA mutation status, receipt of breast-conserving surgery and adjuvant chemotherapy were independently associated with the risk of developing second breast cancers after negative surveillance MRI.

**Data Availability Statement:** All relevant data are within the manuscript and its Supporting Information files.

**Funding:** This work was supported by a new faculty research seed money grant of Yonsei

University College of Medicine for 2022 (2022-32-0074) (VYP) and a National Research Foundation of Korea (NRF) grant funded by the Korea Medical Device Development Fund grant funded by the Korea government (the Ministry of Science and ICT, the Ministry of Trade, Industry and Energy, the Ministry of Health & Welfare, Republic of Korea, the Ministry of Food and Drug Safety) (Project Number: KMDF202011A01-04) (MJK).

**Competing interests:** The authors have declared that no competing interests exist.

**Abbreviations:** PHBC, Personal history of breast cancer; BI-RADS, Breast Imaging Reporting and Data System; DCIS, Ductal carcinoma in situ; IQR, Interquartile range; OR, Odds ratio; CI, Confidence interval.

## Introduction

Breast MRI has been consistently reported to show superior cancer detection rates to mammography or ultrasound (US), thereby enabling early detection of cancer [1,2]. Based on its higher sensitivity in women with dense parenchymal tissue or with post-treatment changes, breast MRI has demonstrated its potential as a supplemental method in several population groups, particularly in women with dense breast tissue or a personal history of breast cancer (PHBC) [2–4]. Accordingly, the 2018 American College of Radiology recommendations now recommend annual surveillance with breast MRI for women with a PHBC and dense breast tissue, or those diagnosed before age 50 years [5].

However, there has been limited literature regarding outcomes following negative surveillance MRI results in women with a PHBC. One recent study reported stable cancer detection rates of 1.6–1.8 per 1000 examinations up to the third surveillance round in women with a PHBC [6]. In contrast, our previous study reported that 90% of subsequent cancers following negative surveillance MRI results were detected at intervals longer than 24 months, in a population in which annual surveillance MRI was not routinely performed [7]. However, all surveillance MRI examinations were performed approximately 2 years after surgery, limiting its application. As recurrence rates peak at two years after diagnosis for HER2 positive and triple negative subtypes and then decrease [8], outcomes regarding subsequent cancers may differ according to interval from surgery.

Therefore, the purpose of this study was to compare outcomes following a negative surveillance MRI result by interval following surgery and investigate factors associated with second breast cancers in women with a PHBC.

## Methods

### Patients

The institutional review board of Severance Hospital approved this retrospective study, and the requirement for informed consent was waived. Since 2013, surveillance breast MRI has been implemented as part of the routine post treatment surveillance protocol at our institution, a tertiary referral university hospital, for patients who have undergone definitive surgery for breast cancer [7]. Surveillance breast MRI is generally recommended at approximately 2 and 5 years after surgery but is also performed following clinician's or patients' requests. We identified consecutive women with a PHBC who underwent prevalence surveillance breast MRI between August 2014 and December 2016. During surveillance, patients are also followed up by using annual mammography and breast US every 6–12 months. Surveillance MRI is usually performed with mammography, whereas US is usually alternatively scheduled at 6–12-month intervals [7].

Inclusion criteria were negative or benign findings (American College of Radiology Breast Imaging Reporting and Data System [BIRADS] category 1, 2) at prevalence surveillance breast MRI. Exclusion criteria were (a) positive findings at surveillance MRI (BI-RADS category 3, 4, or 5), (b) newly detected distant metastasis at the time of surveillance MRI, (c) bilateral mastectomy, and (d) less than 12 months of follow-up after surveillance MRI.

Of the final sample of 1552 women, 993 women had previously been reported [7,9]. Previous studies dealt with outcomes of surveillance MRI, including the cancer detection rate, or dealt with outcomes following surveillance MRI performed at approximately 2 years after surgery. This study reports outcomes following negative surveillance MRI performed at various intervals from breast cancer surgery.

## MRI examination

Breast MRI was performed with two 3-T MRI systems (Ingenia, Philips Medical Systems; Discover 750, GE Medical Systems). MRI protocols consisted of an axial, T2-weighted, fast spin-echo and T2-stimulated inversion recovery sequence (eTHRIVE imaging [Philips Medical Systems], repetition time msec/echo time msec = 3400/80; VIBRANT-Flex dynamic imaging [GE Medical Systems], 5000/70) before contrast material injection and an axial, T1-weighted, dynamic contrast-enhanced sequence with one precontrast and six postcontrast acquisitions (eTHRIVE imaging: matrix = 320 * 410, field of view = 250 * 320 mm, flip angle = 17˚, section thickness = 3 mm, no intersection gap; VIBRANT-Flex dynamic imaging: matrix = 280 * 512, field of view = 320 * 320 mm, flip angle = 12˚, section thickness = 3 mm, no intersection gap) [7].

## MRI evaluation

One of four radiologists with 8–20 years of experience in breast MRI (including V.Y.P., J.H.Y., and M.J.K.) interpreted the MRI examinations, with findings reported accordin to the BI-RADS lexicon [10]. Examinations assessed as BIRADS category 1 or 2 were defined as final negative MRI results and were included in the study. For lesions that were assessed as BIRADS category 4 or 5, targeted US and subsequent US-guided or MRI-guided biopsy was performed accordingly. For lesions assessed as BI-RADS category 3, follow-up imaging at 6 months was recommended.

## Data collection

Age, mammographic breast density, presence of a family history of breast cancer in a first-degree relative, histologic type, clinical TNM stage, estrogen receptor, progesterone receptor, and human epidermal growth factor receptor type 2 status, type of surgery, information about administered treatment (neoadjuvant chemotherapy, adjuvant radiation therapy, adjuvant chemotherapy, and adjuvant endocrine therapy), time from the first breast cancer surgery and information on patient follow-up were collected from the medical records. The first time we accessed the data was on September 1, 2022. The data collection period was until March 31, 2023.

## Statistical analysis

For outcome analysis, second breast cancer was defined as cancer in the ipsilateral breast following breast-conserving surgery or contralateral breast cancer. Local-regional recurrence was defined as recurrence involving the ipsilateral mastectomy bed, ipsilateral axillary lymph nodes, supraclavicular fossa, or infraclavicular and internal mammary lymph nodes (i.e., local-regional recurrence other than those occurring in the ipsilateral breast after breast conserving surgery) [7].

The incidence of second breast cancers and the incidence of second breast cancer or locoregional recurrence were compared based on intervals between initial surgery and negative surveillance breast MRI (less than 3 years vs. 3 years or more).

In order to identify risk factors associated with second breast cancer, the clinical-pathologic characteristics of patients and initial breast cancers were compared on the basis of second breast cancer status. Characteristics were also compared according to second breast cancer or local-regional recurrence status to account for the field of view that is generally covered with breast MRI. Categorical variables were compared by using the chi-square test or the Fisher exact test, and continuous variables were compared by using the Student t test or Mann-

Whitney U test. Multivariable logistic regression analysis was applied to examine the factors associated with outcomes. $p < .05$ was considered to indicate a statistically significant difference.

## Results

### Patient characteristics

Of 1716 women, 164 women were excluded because of MRI results assessed as BI-RADS category 3 (n = 64), BI-RADS category 4 or 5 (n = 14), history of bilateral mastectomy (n = 20), newly detected distant metastasis (n = 4) or no follow-up data more than 12 months (n = 62) (Fig 1). Therefore, 1552 women (median age, 53 years; interquartile range, 46–60 years) were finally included. The original breast cancer diagnosis was invasive cancer in 1327 (85.5%) and ductal carcinoma in situ (DCIS) in 225 (14.5%) patients.

### Overall outcomes after surveillance MRI

Of the 1552 women, 38 (2.4%) developed a second breast cancer (1.6%, 25 of 1552) or local-regional recurrence (0.8%, 13 of 1552) during a median follow-up of 34.9 months after negative surveillance MRI (interquartile range [IQR], 24.8–50.7 months). The 25 second breast cancers (8 ipsilateral breast cancers and 17 contralateral breast cancers), occurred at a median interval of 40.0 months (IQR, 31.3–53.3 months; range, 13.3–73.0 months) from the negative

**Fig 1. Flowchart of study sample.** PHBC = personal history of breast cancer, BI-RADS = Breast Imaging Reporting and Data System, F/U = follow-up.

**Table 1. Patient characteristics according to second breast cancer status.**

| Variable | All patients (n = 1552) | Patients w/o second breast cancer (n = 1527) | Patients with second breast cancer (n = 25) | p value |
|---|---|---|---|---|
| Patient age (y)* | 53 ± 10 | 53 ± 10 | 50 ± 9 | .13 |
| Age at the time of surgery (y) | | | | .25 |
| ≥ 40 | 234 (15.2) | 228 (15.1) | 6 (24.0) | |
| < 40 | 1305 (84.8) | 1286 (84.9) | 19 (76.0) | |
| Mammographic breast density | | | | .41 |
| BI-RADS A or B | 741 (47.7) | 727 (47.6) | 14 (56.0) | |
| BI-RADS C or D | 811 (52.3) | 800 (52.4) | 11 (44.0) | |
| Family history of breast cancer | | | | .09 |
| No | 1443 (93.0) | 1422 (93.1) | 21 (84.0) | |
| Yes | 109 (7.0) | 105 (6.9) | 4 (16.0) | |
| Original breast cancer histologic type | | | | .24 |
| Invasive | 1327 (85.5) | 1308 (85.7) | 19 (76.0) | |
| Ductal carcinoma in situ | 225 (14.5) | 219 (14.3) | 6 (24.0) | |
| TNM stage | | | | .29 |
| 0 | 225 (14.6) | 219 (14.4) | 6 (24.0) | |
| 1 | 661 (42.8) | 650 (42.7) | 11 (44.0) | |
| 2 | 532 (34.4) | 524 (34.5) | 8 (32.0) | |
| 3 | 128 (8.3) | 128 (8.4) | 0 (0.0) | |
| Type of surgery | | | | .01 |
| Breast conserving | 951 (61.3) | 929 (60.8) | 22 (88.0) | |
| Mastectomy | 601 (38.7) | 598 (39.2) | 3 (12.0) | |
| Neoadjuvant chemotherapy | | | | .41 |
| No | 1298 (83.6) | 1275 (83.5) | 23 (92.0) | |
| Yes | 254 (16.4) | 252 (16.5) | 2 (8.0) | |
| Adjuvant radiation therapy | | | | .12 |
| No | 539 (34.7) | 534 (35.0) | 5 (20.0) | |
| Yes | 1013 (65.3) | 993 (65.0) | 20 (80.0) | |
| Adjuvant chemotherapy | | | | .02 |
| No | 1083 (69.8) | 1071 (70.1) | 12 (48.0) | |
| Yes | 469 (30.2) | 456 (29.9) | 13 (52.0) | |
| Antihormonal therapy | | | | .77 |
| No | 336 (21.6) | 330 (21.6) | 6 (24.0) | |
| Yes | 1216 (78.4) | 1197 (78.4) | 19 (76.0) | |

Note.—Unless otherwise noted, data are numbers of patients, with percentages in parentheses. BI-RADS = Breast Imaging Reporting and Data System; w/o = without.

* Data are means ± standard deviations.

surveillance MRI study. None had concurrent local-regional recurrence or distant metastasis. Table 1 lists the baseline characteristics of the study population according to outcome.

Of the 25 second breast cancers, 23 (92%) were DCIS (n = 7) or node-negative T1 cancers (n = 16). In two patients, a clinically node-positive T1 cancer was detected and pathologic complete response was achieved after neoadjuvant chemotherapy. Of the 25 second breast cancers, only two (8%) were detected less than 24 months after the negative surveillance MRI study: a node-negative T1mi cancer and a low-grade ductal carcinoma in situ (DCIS). Eleven of the 25 second breast cancers (44%) were mammographically occult. Notably, 6 (24%) were detected only with subsequent surveillance breast MRI performed at a median of 33.2 months (range, 28.1–55.5 months) after a negative MRI, although mammography had been performed

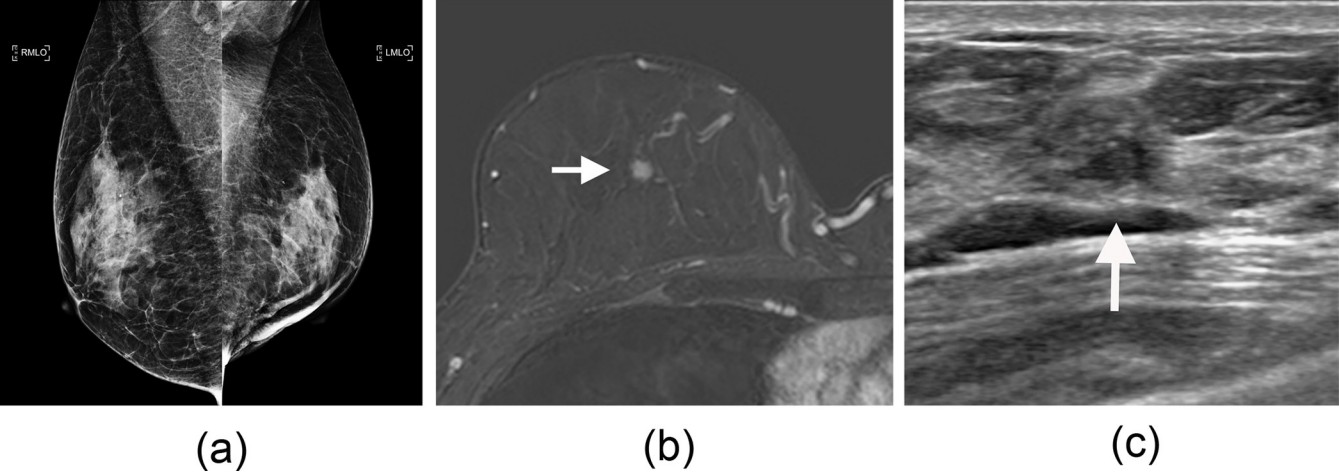

**Fig 2. Images of a second breast cancer detected in a 53-year-old woman on subsequent breast MRI.** She had undergone breast-conserving surgery of her left breast followed by radiation and antihormonal therapy for a grade 2 invasive ductal carcinoma (hormone receptor-positive, human epidermal growth factor receptor type 2-positive, pT1N0). Her previous prevalence surveillance breast MRI, obtained within 3 years following surgery, was assessed as negative for cancer. (a) Mammogram shows dense breast tissue and was assessed as negative for cancer. (b) Axial contrast-enhanced T1-weighted MRI scan obtained 33.1 months after her negative breast MRI result shows a newly developed 9-mm mass in her right upper central breast (arrow). (c) Targeted US shows a hypoechoic mass (arrow) which was confirmed as DCIS. LMLO = left mediolateral oblique, RMLO = right mediolateral oblique/.

at a median of 10.5 months (range, 4.9–13.9 months) before the MRI examinations (Fig 2 and S1 Table).

The 13 local-regional recurrences occurred at a median interval of 18.9 months (range, 6.2–66.2 months) from the negative surveillance MRI (S1 Appendix).

## Comparison of second breast cancers according to intervals ($<$3 years vs $\geq$ 3 years) following initial surgery

The median interval between initial surgery and surveillance MRI were 26.1 months (IQR, 24.2–31.5; surgery-MRI interval $<$ 3 years group) and 55.6 months (IQR, 43.0–57.3; surgery-MRI interval $\geq$ 3 years group), respectively. The incidence of second breast cancers (1.4% [14/1021] vs 2.1% [11/531], $p = .30$) or that of second breast cancers or local-regional recurrence (2.5% [25/1021] vs 2.5% [13/531], $p > .99$) did not significantly differ by surgery-MRI interval ($<$3 years vs $\geq$ 3 years) (Table 2). The median follow-up period for women without recurrence were 57.4 (IQR, 46.5–65.6) and 54.4 (IQR, 48.7–64.4) months for each group, respectively.

There was no significant difference in intervals between negative surveillance MRI and detection of second breast cancer (34.9 [IQR, 28.4–47.2] vs 50.6 [IQR, 32.7–55.5] months, $p = .202$) or intervals between negative surveillance MRI and detection of second breast cancer or local-regional occurrence (42.4 [IQR, 32.2–53.7] vs 33.1 [IQR, 18.0–42.4], $p = .110$) between the two groups (surgery-MRI interval $<$3 years vs $\geq$ 3 years). Proportions of node negative T1 or DCIS did not significantly differ- between the two interval groups (85.7% [12 of 14] vs 100% [11 of 11], $p = .487$). However, all the two node-positive invasive second breast cancers occurred in the group with a negative MRI performed $<$ 3 years following initial surgery, although US had been performed 8 months and 12 months before detection, respectively.

## Factors associated with second breast cancers

To identify factors associated with second breast cancer, variables showing $p$ values less than .05 in univariable logistic regression analysis, including BRCA mutation status (odds ratio

**Table 2. Comparison of second breast cancer characteristics according to initial surgery-MRI intervals.**

| Outcome | Initial surgery-MR interval <3 years | Initial surgery-MR interval ≥ 3 years | p value |
|---|---|---|---|
| Second breast cancer rate* | 13.7 (14/1021) | 20.7 (11/531) | .30 |
| Second breast cancer or LR rate* | 24.5 (25/1021) | 24.5 (13/531) | >.99 |
| Characteristics of second breast cancers | | | |
| MR-detection interval† | 34.9 (28.4, 47.2) | 50.6 (32.7, 55.5) | .20 |
| Percentage of node negative invasive cancers (%) | 64% (9/14) | 64% (7/11) | >.99 |
| Percentage of DCIS (%) | 21% (3/14) | 36% (4/11) | .35 |
| Percentage of node-negative T1 cancer or DCIS (%) | 86% (12/14) | 100% (11/11) | .49 |

Note.—Except where indicated, numbers in parentheses are raw data.

* Cancer rate is defined as the number of cancers detected per 1000 examinations.

† Data are median intervals in months. Numbers in parentheses are interquartile range.

LR = Local-regional recurrence, DCIS = Ductal carcinoma in situ.

[OR], 8.92; 95% confidence interval [CI]: 2.49–31.90]), receipt of breast-conserving surgery (OR, 4.72; 95% CI: 1.41–15.84), and receipt of adjuvant chemotherapy (OR, 2.54; 95% CI: 1.15–5.62), were selected as input variables for multivariable analysis. At multivariable logistic regression analysis, BRCA mutation status (OR, 10.35; 95% CI: 2.74, 39.07; $p$ = .001), receipt of breast-conserving surgery (OR, 5.68; 95% CI: 1.66, 19.51; $p$ = .006), and adjuvant chemotherapy (OR, 2.70; 95% CI: 1.21, 6.03; $p$ = .016) remained as significant factors (Table 3).

When investigating factors associated with second breast cancer or locoregional recurrence, only receipt of adjuvant chemotherapy (OR, 2.10; 95% CI: 1.08–4.09) showed an association according to univariable logistic regression analysis (S2 Table).

## Discussion

With the increase in number of breast cancer survivors and no definite upper age limit for surveillance, information regarding outcomes according to interval after initial breast cancer surgery would be helpful in determining practical supplemental surveillance strategies. Based on 1552 consecutive women who underwent prevalence surveillance breast MRI at different intervals following initial surgery, we found that there was no significant difference in outcomes following negative surveillance MRI by interval following surgery (< 3 years vs ≥ 3 years). Therefore, at least for most patients, differing surveillance breast MRI intervals according to period from initial surgery does not seem necessary.

Similar to our previous report [7], we found that only two second breast cancers were detected within 24 months following a negative surveillance MRI. Although a previous retrospective study reported stable cancer detection rates for annual surveillance breast MRI in women who underwent breast-conserving surgery, a significant proportion (> 30%) of patients had dropped out at each subsequent round, possibly resulting in higher-risk patients at later surveillance rounds [6,11]. In contrast, cancer detection rates have been reported to decrease in subsequent rounds in a multi-center prospective study [2]. The recent DENSE trial has also shown similar results, reporting lower cancer detections rates in the second surveillance round in women with extremely dense breasts [12]. In addition, all cancers detected at the second round in the aforementioned studies were stage 0 or stage 1 cancers [2,6,12]. The European Society of Breast Imaging now recommends supplemental MRI surveillance to be offered for women with extremely dense breasts at least every 4

**Table 3. Factors associated with second breast cancers.**

| Variable | Univariable analysis | | Multivariable analysis | |
|---|---|---|---|---|
| | Odds ratio | *p* value | Odds ratio | *p* value |
| Age at surgery (y) | | .22 | | |
| < 40 | 1.78 (0.70, 4.51) | | | |
| ≥ 40 | Reference | | | |
| Mammographic breast density | | .41 | | |
| BI-RADS A or B, fatty | Reference | | | |
| BI-RADS C or D, dense | 0.71 (0.32, 1.58) | | | |
| BRCA mutation status | | .001 | | .001 |
| Negative | Reference | | | |
| Positive | 8.92 (2.49, 31.90) | | 10.35 (2.74, 39.07) | |
| First-degree family history of breast cancer | | .09 | | |
| No | Reference | | | |
| Yes | 2.58 (0.87, 7.65) | | | |
| Stage of primary breast cancer | | .18 | | |
| Ductal carcinoma in situ | 1.89 (0.75, 4.78) | | | |
| Invasive | Reference | | | |
| TNM stage | | .74 | | |
| 0 | Reference | | | |
| 1 | 0.62 (0.23, 1.69) | | | |
| 2 | 0.56 (0.19, 1.63) | | | |
| ER and PR status | | .62 | | |
| ER- and PR-negative | 0.76 (0.25, 2.25) | | | |
| ER- or PR-positive | Reference | | | |
| HER2 status | | .68 | | |
| Negative | Reference | | | |
| Positive | 1.25 (0.45, 3.49) | | | |
| Type of surgery | | .012 | | .006 |
| Breast conserving | 4.72 (1.41, 15.84) | | 5.68 (1.66, 19.51) | |
| Mastectomy | Reference | | | |
| Adjuvant radiation therapy | | .13 | | |
| No | 0.47 (0.17, 1.25) | | | |
| Yes | Reference | | | |
| Neoadjuvant chemotherapy | | .27 | | |
| No | Reference | | | |
| Yes | 0.44 (0.10, 1.88) | | | |
| Adjuvant chemotherapy | | .021 | | .016 |
| No | Reference | | | |
| Yes | 2.54 (1.15, 5.62) | | 2.70 (1.21, 6.03) | |
| Antihormonal therapy | | .77 | | |
| No | 1.15 (0.45, 2.89) | | | |
| Yes | Reference | | | |

Note.—Data in parentheses are 95% CIs. ER = estrogen receptor, HER2 = human.

epidermal growth factor receptor type 2, NA = not applicable, PR = progesterone receptor.

years, preferably every 2 to 3 years [13]. Our results are similar in that few second cancers were detected within 24 months in women with PHBC, most of who are of intermediate life-time risk. However, as our study sample was comprised of Asian women, outcomes may

differ in corresponding subgroups of Western populations as population-based breast cancer incidence rates are higher [14].

The risk of local regional recurrence as a first event after breast cancer treatment decreases with increasing event-free years. The reported risk of local recurrence as a first event before 5 years was 2.4% and 0.6% after two and four event-free years, respectively, in a recent study [15]. However, we found no significant difference in the incidence of second breast cancers or that of second breast cancers or local-regional recurrence following negative surveillance breast MRI between the two groups (surgery-MRI interval <3 years vs ≥ 3 years). This is likely because contralateral breast cancers accounted for 68% of second breast cancers, which risk is relatively constant regardless of the period from initial diagnosis. However, we did find that the two node-positive second breast cancers all occurred in women with negative MRI examinations performed < 3 years after initial surgery. Although the numbers are small, this suggests that a small number of women would likely benefit from more frequent surveillance MRI in the first five years after diagnosis. As annual surveillance MRI may not be feasible in many institutions, efforts to further improve risk stratification are needed. Nonetheless, longer imaging intervals for supplemental surveillance MRI may still be appropriate for most patients, especially for those without other high risk factors, as the overall second breast cancer rate following a negative surveillance breast MRI was low (1.6%, 25 of 1552).

Although there was no significant difference in intervals between surveillance MRI and detection of second breast cancer between the two groups, the interval seemed longer (34.9 vs. 50.6) in women with a negative MRI ≥ 3 years following initial surgery. This is likely attributed to the fact that most women would not have undergone subsequent surveillance MRI in this subgroup, as MRI is partially covered by the Korea National Health Insurance Service for only up 5 years after surgery. Yet, all second breast cancers in this group were node-negative T1 or DCIS, suggesting that more frequent MRI surveillance may not improve outcomes in most patients after the first 5 years after diagnosis.

Similar to previous studies, we also found that the presence of BRCA mutation, receipt of breast-conserving surgery and adjuvant chemotherapy were associated with second breast cancer risk [16,17]. As adjuvant chemotherapy is administered in patients at higher risk of recurrence, this variable also showed association with second breast cancer or locoregional recurrence. Our results are in line with previous studies and imply that these factors should be considered when establishing personalized imaging surveillance strategies.

Our study had several limitations. First, this was a retrospective study conducted at a single institution. Second, although we included women who underwent surveillance MRI at different intervals from surgery, the time span of negative MRI examinations were all within the first five years of diagnosis. However, as the median follow-up for women without recurrence was 56.0 months, we were able to follow-up most women for up to 10 years after initial cancer diagnosis. Finally, our study sample was comprised of Asian women, who have lower breast cancer incidence rates than Western populations.

In summary, outcomes following a negative surveillance MRI did not differ by surgery-MRI interval and 92% of second breast cancers were early stage cancers. BRCA mutation status, receipt of breast-conserving surgery and adjuvant chemotherapy were independently associated with the risk of developing second breast cancers after negative surveillance MRI. These observations may help when planning subsequent surveillance MRI for women with a PHBC.

## Supporting information

**S1 Appendix. Site of local-regional recurrences.**
(DOCX)

**S1 Table. Clinical-pathologic imaging characteristics of the 25 second breast cancers.**
(DOCX)

**S2 Table. Factors associated with second breast cancers or locoregional recurrence.**
(DOCX)

**S1 Data.**
(XLSX)

## Author Contributions

**Conceptualization:** Vivian Youngjean Park.

**Data curation:** Yohan Joo, Min Jung Kim, Jung Hyun Yoon, Miribi Rho, Vivian Youngjean Park.

**Formal analysis:** Yohan Joo, Vivian Youngjean Park.

**Funding acquisition:** Vivian Youngjean Park.

**Investigation:** Yohan Joo.

**Methodology:** Vivian Youngjean Park.

**Writing – original draft:** Yohan Joo.

**Writing – review & editing:** Yohan Joo, Min Jung Kim, Jung Hyun Yoon, Miribi Rho, Vivian Youngjean Park.

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
