## [Decision Letter · Decision Letter 0]

8 Apr 2024

PONE-D-24-00795Second breast cancer following negative breast MRI: analysis by interval from surgery and risk factorsPLOS ONE

Dear Dr. Park,

Thank you for submitting your manuscript to PLOS ONE. After careful consideration, we feel that it has merit but does not fully meet PLOS ONE’s publication criteria as it currently stands. Therefore, we invite you to submit a revised version of the manuscript that addresses the points raised during the review process.

We look forward to receiving your revised manuscript.

Kind regards,

Daniele Ugo Tari, M.D.

Academic Editor

PLOS ONE

Outcomes Following Negative Screening MRI Results in Korean Women with a Personal History of Breast Cancer: Implications for the Next MRI Interval - https://doi.org/10.1148/radiol.2021204217

In your revision ensure you cite all your sources (including your own works), and quote or rephrase any duplicated text outside the methods section. Further consideration is dependent on these concerns being addressed.

Reviewers' comments:

Reviewer's Responses to Questions

**Comments to the Author**

1. Is the manuscript technically sound, and do the data support the conclusions?

Reviewer #1: Yes

Reviewer #2: Yes

2. Has the statistical analysis been performed appropriately and rigorously? 

Reviewer #1: Yes

Reviewer #2: Yes

3. Have the authors made all data underlying the findings in their manuscript fully available?

Reviewer #1: Yes

Reviewer #2: Yes

4. Is the manuscript presented in an intelligible fashion and written in standard English?

Reviewer #1: Yes

Reviewer #2: Yes

5. Review Comments to the Author

Reviewer #1: Although it did not have any effect on my acceptance when I evaluated the current article, patients receiving neoadjuvant chemotherapy could have been excluded to ensure homogeneity when selecting the patients in the sample. I don't say that this should have been done because it would not affect its contribution to the literature when compared to similar articles in the literature. The subject is an area of my interest and in which I follow the literature, and it is written carefully and the data results are well-examined in the discussion. Therefore, I think there is no need for revision in the article.

Reviewer #2: This retrospective study on second breast cancers following negative breast MRI examinations is well written and practical.

1. In Abstract Objectives, the first sentence needs clarification.

2. In Abstract Results, the second sentence needs re-writing. Second breast cancers is duplicated in that sentence. In that sentence, I suggest starting the sentence with The incidence. In the third sentence, I suggest starting the sentence with The median intervals. In the third sentence and throughout the paper, the two groups should be clearly defined. In the last sentence of Abstract Results, the authors should indicate that BRCA mutations, breast conservation therapy, and adjuvant chemotherapy were significant factors for developing a second breast cancer.

3. In Introduction, 1st paragraph, last sentence, years should be added after 50.

4. In Methods, MRI evaluation, 1st sentence, lexicon should be added after BI-RADS. In the last sentence, I suggest changing imaging to MRI.

5. In Methods, Statistical analysis, the second paragraph needs clarification and the sentence should start with The incidence.

6. In Results, Patient characteristics, last sentence, I suggest: and ductal carcinoma in situ (DCIS) in 225 (14.5%) patients.

7. In Results, Overall outcomes after surveillance MRI, 1st paragraph, 1st sentence, I suggest adding a before second

8. In Results, Overall outcomes after surveillance MRI, 2nd paragraph, last sentence, I suggest: after a negative MRI...before the MRI examinations.

9. In Results, Comparison of second breast cancers according to intervals (<3 years vs >3 years) following initial surgery, 1st and 2nd paragraphs, 1st sentence, each group should be defined.

10. In Discussion, 2nd paragraph, 1st sentence, I suggest: only two breast cancers were.

11. In Discussion, 3rd paragraph, 3rd sentence, the should be added before incidence and the two groups should be defined. In the 4th sentence, the should be added between of and period. In the last sentence of that paragraph, factor should be changed to factors.

12. The authors should check the journal's reference style. In references 3, 4, 8, and 12, I think that the first 3 authors should be listed, followed by et al. In references 6, 7, 11-13, 15, and 16, only the first word of the title of the article should be capitalized.

13. I suggest that the authors add a figure showing a second breast cancer identified on MRI.

6. PLOS authors have the option to publish the peer review history of their article (what does this mean?). If published, this will include your full peer review and any attached files.

Reviewer #1: **Yes: **ONUR DÜLGEROĞLU

Reviewer #2: No

---

## [Author Response · Author response to Decision Letter 0]

11 Apr 2024

Response to Comments from the Reviewers;

Reviewer #1: Although it did not have any effect on my acceptance when I evaluated the current article, patients receiving neoadjuvant chemotherapy could have been excluded to ensure homogeneity when selecting the patients in the sample. I don't say that this should have been done because it would not affect its contribution to the literature when compared to similar articles in the literature. The subject is an area of my interest and in which I follow the literature, and it is written carefully and the data results are well-examined in the discussion. Therefore, I think there is no need for revision in the article.

: Response – Thank you for your kind comment. In our study, we did not exclude patients who underwent neoadjuvant chemotherapy, which comprised 16.4% (254 of 1552) of our study sample (Table 1). We agree that this would not have affected our outcomes, as there was no significant difference by surgery-MRI interval. 

Reviewer #2: This retrospective study on second breast cancers following negative breast MRI examinations is well written and practical.

1. In Abstract Objectives, the first sentence needs clarification.

: Response – Thank you for your comment. In our initially submitted manuscript, we had included the first sentence to provide some background on why we performed this study. As the risk of local regional recurrence as a first event after breast cancer treatment decreases with increasing event-free years, outcomes following negative surveillance breast MRI results may differ according to interval from initial surgery. Therefore, the main objective of this study was to compare outcomes following a negative surveillance MRI study by surgery-MRI interval. 

 Upon re-reviewing the manuscript, we acknowledge that this sentence can be confusing in the Abstract and is unnecessary. Therefore, we have deleted it in our revised manuscript. 

2. In Abstract Results, the second sentence needs re-writing. Second breast cancers is duplicated in that sentence. In that sentence, I suggest starting the sentence with The incidence. In the third sentence, I suggest starting the sentence with The median intervals. In the third sentence and throughout the paper, the two groups should be clearly defined. In the last sentence of Abstract Results, the authors should indicate that BRCA mutations, breast conservation therapy, and adjuvant chemotherapy were significant factors for developing a second breast cancer.

: Response – Thank you for your helpful comment. We have revised the Abstract Results section accordingly 

(Amended text)

• Abstract Results: “The incidence of second breast cancers or local-regional recurrence did not significantly differ by surgery-MRI interval. The median intervals between MRI to second breast cancer detection showed no significant difference between the two groups (surgery-MRI interval <3 years vs. ≥ 3 years). Two node-positive second breast cancers were detected in the group with <3 years interval. BRCA mutation status, receipt of breast-conserving surgery, and adjuvant chemotherapy (all p < .05) were significant factors associated with the development of second breast cancers”, Page 2 in revised clean manuscript.

3. In Introduction, 1st paragraph, last sentence, years should be added after 50.

: Response – Thank you for your helpful comment. We have revised the sentence accordingly. 

(Amended text)

• Introduction: “Accordingly, the 2018 American College of Radiology recommendations now recommend annual surveillance with breast MRI for women with a PHBC and dense breast tissue, or those diagnosed before age 50 years [5]”, Page 4 in revised clean manuscript. 

4. In Methods, MRI evaluation, 1st sentence, lexicon should be added after BI-RADS. In the last sentence, I suggest changing imaging to MRI.

: Response – Thank you for your comment. We have revised the first sentence accordingly. Although follow-up MRI is primarily recommended for BI-RADS category 3 lesions, targeted ultrasound or follow-up ultrasound at 6 months is also performed at our institution. Physicians sometimes opt to order an additional MRI when there is no US correlate. Therefore, we did not change our last sentence. However, we are willing to make changes if necessary in future revisions, as this sentence regards recommendations. 

(Amended text)

• Methods, MRI Evaluation: “One of four radiologists with 8-20 years of experience in breast MRI (including V.Y.P., J.H.Y., and M.J.K.) interpreted the MRI examinations, with findings reported according to the BI-RADS lexicon”, Page 6 of revised clean manuscript.

5. In Methods, Statistical analysis, the second paragraph needs clarification and the sentence should start with The incidence.

: Response – Thank you for your comment. We have revised the sentence accordingly, and have added ‘the incidence’ in the middle of the sentence to make it clearer. 

(Amended text)

• Methods, Statistical Analysis: “The incidence of second breast cancers and the incidence of second breast cancer or locoregional recurrence were compared based on intervals between initial surgery and negative surveillance breast MRI (less than 3 years vs. 3 years or more)”, Page 7 of revised clean manuscript. 

6. In Results, Patient characteristics, last sentence, I suggest: and ductal carcinoma in situ (DCIS) in 225 (14.5%) patients.

: Response – Thank you for your helpful comment. 

(Amended text)

• Results, Patient Characteristics: “The original breast cancer diagnosis was invasive cancer in 1327 (85.5%) and ductal carcinoma in situ (DCIS) in 225 (14.5%) patients”, Page 8 of revised clean manuscript. 

7. In Results, Overall outcomes after surveillance MRI, 1st paragraph, 1st sentence, I suggest adding a before second

: Response – Thank you for your helpful comment. 

(Amended text)

• Results, Overall outcomes after surveillance MRI: “Of the 1552 women, 38 (2.4%) developed a second breast cancer (1.6%, 25 of 1552) or local-regional recurrence (0.8%, 13 of 1552) during a median follow-up of 34.9 months after negative surveillance MRI (interquartile range [IQR], 24.8–50.7 months)”, Page 8 of revised clean manuscript.

8. In Results, Overall outcomes after surveillance MRI, 2nd paragraph, last sentence, I suggest: after a negative MRI...before the MRI examinations.

: Response – Thank you for your helpful comment. We have revised the sentence accordingly.

(Amended text)

• Results, Overall outcomes after surveillance MRI: “Notably, 6 (24%) were detected only with subsequent surveillance breast MRI performed at a median of 33.2 months (range, 28.1–55.5 months) after a negative MRI, although mammography had been performed at a median of 10.5 months (range, 4.9–13.9 months) before the MRI examinations (Fig 2 and S1 Table)”, Page 10 of revised clean manuscript.

9. In Results, Comparison of second breast cancers according to intervals (<3 years vs >3 years) following initial surgery, 1st and 2nd paragraphs, 1st sentence, each group should be defined.

: Response – Thank you for your helpful comment. We have clearly defined each group in the first sentence of both paragraphs in our revised manuscript. 

(Amended text)

• Results, Comparison of second breast cancers according to intervals (<3 years vs ≥ 3 years) following initial surgery: 

“The median interval between initial surgery and surveillance MRI were 26.1 months (IQR, 24.2–31.5; surgery-MRI interval < 3 years group) and 55.6 months (IQR, 43.0–57.3; surgery-MRI interval ≥ 3 years group), respectively”, Page 11 of revised clean manuscript. 

“There was no significant difference in intervals between negative surveillance MRI and detection of second breast cancer (34.9 [IQR, 28.4–47.2] vs 50.6 [IQR, 32.7–55.5] months, p = .202) or intervals between negative surveillance MRI and detection of second breast cancer or local-regional occurrence (42.4 [IQR, 32.2–53.7] vs 33.1 [IQR, 18.0–42.4], p = .110) between the two groups (surgery-MRI interval <3 years vs ≥ 3 years)”, Page 12 of revised clean manuscript.

10. In Discussion, 2nd paragraph, 1st sentence, I suggest: only two breast cancers were.

: Response – Thank you for your helpful comment. We have revised the above sentence. 

(Amended text)

• Discussion, 2nd paragraph: “Similar to our previous report [7], we found that only two second breast cancers were detected within 24 months following a negative surveillance MRI”. Page 14 of revised clean manuscript.

11. In Discussion, 3rd paragraph, 3rd sentence, the should be added before incidence and the two groups should be defined. In the 4th sentence, the should be added between of and period. In the last sentence of that paragraph, factor should be changed to factors.

: Response – Thank you for your helpful comment. We have revised the sentences accordingly.

(Amended text)

• Discussion, 3rd paragraph: “However, we found no significant difference in the incidence of second breast cancers or that of second breast cancers or local-regional recurrence following negative surveillance breast MRI between the two groups (surgery-MRI interval <3 years vs ≥ 3 years). This is likely because contralateral breast cancers accounted for 68% of second breast cancers, which risk is relatively constant regardless of the period from initial diagnosis.”, Page 15 in revised clean manuscript. 

“Nonetheless, longer imaging intervals for supplemental surveillance MRI may still be appropriate for most patients, especially for those without other high risk factors, as the overall second breast cancer rate following a negative surveillance breast MRI was low (1.6%, 25 of 1552)”, Page 15 in revised clean manuscript. 

12. The authors should check the journal's reference style. In references 3, 4, 8, and 12, I think that the first 3 authors should be listed, followed by et al. In references 6, 7, 11-13, 15, and 16, only the first word of the title of the article should be capitalized.

: Response – Thank you for your thorough comment. We have rechecked the submission guidelines for PLOS ONE, and the references guidelines state that “References with more than six authors should list the first six author names, followed by “et al.” Therefore, we did not change references 3, 4, 8 and 12. We failed to find specific guidelines regarding capitalization of letters, but have changed references 6, 7, 11-13 accordingly – only the first word of the title of the article has been capitalized, with the exception of abbreviations (MRI) and proper nouns. 

13. I suggest that the authors add a figure showing a second breast cancer identified on MRI. 

: Response – Thank you for your helpful comment. In our revised manuscript, we have added Fig 2 showing a second breast cancer case identified on subsequent surveillance MRI.

---

## [Editor Report · Decision Letter 1]

24 Jun 2024

Second breast cancer following negative breast MRI: analysis by interval from surgery and risk factors

PONE-D-24-00795R1

Dear Dr. Park,

We’re pleased to inform you that your manuscript has been judged scientifically suitable for publication and will be formally accepted for publication once it meets all outstanding technical requirements.

Kind regards,

Daniele Ugo Tari, M.D.

Academic Editor

PLOS ONE
---

## [Editor Report · Acceptance letter]

6 Aug 2024

PONE-D-24-00795R1 

PLOS ONE

Dear Dr. Park, 

I'm pleased to inform you that your manuscript has been deemed suitable for publication in PLOS ONE. Congratulations! Your manuscript is now being handed over to our production team.

Kind regards, 

on behalf of

Dr. Daniele Ugo Tari 

Academic Editor

PLOS ONE